

# Preparation, thermal stability and electrical transport properties of vaesite, NiS$_2$

Helena M. Ferreira[1], Elsa B. Lopes[1], José F. Malta[1,2], Luís M. Ferreira[1], Maria H. Casimiro[1], Luís Santos[3], Manuel F.C. Pereira[4] and Antonio Pereira Gonçalves[1]

[1] C2TN, Departamento de Engenharia e Ciências Nucleares, Instituto Superior Técnico, Universidade de Lisboa, Bobadela, Portugal
[2] CFisUC, Departamento de Física, Universidade de Coimbra, Coimbra, Portugal
[3] CQE, Departamento de Engenharia Química, Instituto Superior Técnico, Universidade de Lisboa, Lisboa, Portugal
[4] CERENA, Departamento de Engenharia Civil Arquitectura e Georrecursos, Instituto Superior Técnico, Universidade de Lisboa, Lisboa, Portugal

## ABSTRACT

Vaesite, a nickel chalcogenide with NiS$_2$ formula, has been synthetized and studied by theoretical and experimental methods. NiS$_2$ was prepared by solid-state reaction under vacuum and densified by hot-pressing, at different consolidation conditions. Dense single-phase pellets (relative densities >94%) were obtained, without significant lattice distortions for different hot-pressing conditions. The thermal stability of NiS$_2$ was studied by thermogravimetric analysis. Both as-synthetized and hot-pressed NiS$_2$ have a single phase nature, although some hot-pressed samples had traces of the sulfur deficient phase, Ni$_{1-x}$S (<1%vol), due to the strong desulfurization at T > 340 °C. The electronic band structure and density of states were calculated by Density Functional Theory (DFT), indicating a metallic behavior. However, the electronic transport measurements showed p-type semiconductivity for bulk NiS$_2$, verifying its characteristic behavior has a Mott insulator. The consolidation conditions strongly influence the electronic properties, with the best room-temperature Seebeck coefficient, electrical resistivity and power factor being 182 $\mu VK^{-1}$, 2,257 $\mu \Omega$ m and 14.1 $\mu WK^{-2}$ m$^{-1}$, respectively, pointing this compound as a good starting point for a new family of thermoelectric materials.

## INTRODUCTION

The search for new, clean, energy sources, as well as the optimization of their use, has become a major issue in contemporary societies. According to the European Environment Agency, current conventional thermal power plants have an energy efficiency around 35–45%, most of the energy being lost as wasted heat (*European Environment Agency, 2013*). Thermoelectric (TE) materials, which convert thermal energy into electric energy (Seebeck effect) and vice-versa (Peltier effect), are a promising solution to increase the

Corresponding author
Antonio Pereira Gonçalves,
apg@ctn.tecnico.ulisboa.pt

efficiency of many devices and equipment. The potential of a material for thermoelectrics can be evaluated by its figure of merit, $zT = \alpha^2 T / \rho \lambda$, where $\alpha$, T, $\rho$ and $\lambda$ are the Seebeck coefficient, absolute temperature, electrical resistivity and thermal conductivity, respectively (*Gonçalves & Godart, 2014*).

Current commercially available TE materials contain rare, expensive and toxic elements, being necessary to develop new, cheap, abundant and environment-friendly alternatives. Metal sulfides are interesting candidates, as they fulfill these requirements (*Ge et al., 2016*). Tetrahedrites are cheap and easily available mineral sulfosalts that present large figures of merit and are seen as having good potential for thermoelectric applications (*Lu & Morelli, 2016*). Pyrite ($FeS_2$) is low cost sulfide with simple synthesis and moderate thermoelectric properties (*Harada, 1998*; *Zuñiga Puelles et al., 2019*). In this compound, the large electrical resistivity and thermal conductivity observed in the pristine material are the major constraints to their practical use. These properties can be tuned to much lower values by changing both the composition and microstructure (*Uhlig et al., 2014*). Vaesite ($NiS_2$), another transition mineral sulfide with pyrite structure (*Krill et al., 1976*), was reported, but its thermoelectric properties were only poorly explored. At equilibrium conditions, $NiS_2$ is a stoichiometric compound, stable up to 1,020 °C (*Waldner & Pelton, 2004*). However, previous studies also suggested that vaesite is an intrinsic non-stoichiometric compound, with a variable metal concentration and a stable anion content. These deviations from stoichiometry, corroborated by a change of the cell parameters, have important consequences in the electrical and magnetic properties (*Gautier et al., 1972*; *Krill et al., 1976*). $NiS_2$ was reported to order antiferromagnetically below $T_N$ ~50 K, which is followed by a spin reorientation at ~30 K that leads to a week ferromagnetic ground state (*Yao et al., 1996*). Measurements on single crystals, natural materials and samples prepared by high-pressure synthesis indicated a semiconducting behavior for this compound (*Bither et al., 1968*; *Kautz et al., 1972*; *Gautier et al., 1972*; *Krill et al., 1976*), which pointed to the possibility of using it as thermoelectric material. Nevertheless, thermoelectric measurements made on thin films showed a p-type semiconducting behavior, but small room temperature Seebeck coefficients (4.5–14 µV/K), which contrasts with the large values obtained on single crystals prepared by halogen transport (311–400 µV/K) (*Bither et al., 1968*; *Krill et al., 1976*; *Gautier et al., 1972*; *Ferrer & Sanchez, 1999*; *Clamagirand et al., 2012*; *Kautz et al., 1972*; *Kwizera, Dresselhaus & Adler, 1980*; *Matsuura et al., 2000*). Reference values of Seebeck coefficient, resistivity and power factor of vaesite at room temperature can be found in Table 1. Moreover, albeit the preparation of synthetic bulk $NiS_2$ by solid state route has been previously described (*Krill et al., 1976*; *Matsuura et al., 2000*), it resulted in highly porous pellets, easily disaggregated, not suitable for the electrical transport properties study. In this work, we explored the solid-state route followed by hot-press to prepare dense vaesite samples, suitable for their characterization, including the electrical transport properties (electrical resistivity and Seebeck coefficient) investigation. Density functional theory calculations were also performed and their results compared with the experimental data.

**Table 1** Reference values of Seebeck coefficient, electrical resistivity and power factor of vaesite at room temperature.

| Type of sample | $\alpha$ ($\mu$V/K) | $\rho$ ($\mu\Omega$ m) | Power factor ($\mu$WK$^{-2}$ m$^{-1}$) | Reference |
|---|---|---|---|---|
| Polycrystalline | – | 1,000–30,000 | – | *Krill et al. (1976)* |
| Polycrystalline | – | 1,000–10,000 | – | *Gautier et al. (1972)* |
| Thin film | 4.5–6.3 | 100–180 | – | *Ferrer & Sanchez (1999)* |
| Thin film | 14 | 1,700 | 0.12 | *Clamagirand et al. (2012)* |
| Single crystal | 311–400 | 6,000–30,000 | 5–16 | *Kautz et al. (1972)*, *Kwizera, Dresselhaus & Adler (1980)*, *Matsuura et al. (2000)*, *Bither et al. (1968)* |

## MATERIALS & METHODS

$NiS_2$ samples, with an average mass of $\sim$1.5 g, were prepared by high-temperature reacting the elements inside quartz ampoules. The desired quantities of Ni and S were put inside the quartz ampoules (eight mm inner diameter, one mm wall thickness), which were evacuated down to $6 \times 10^{-5}$ mbar and sealed. An excess of 5 wt% of the chalcogenide element was considered in order to compensate eventual evaporation losses. The ampoules were placed in a horizontal tube furnace pre-heated at 150 °C, and heated at 800 °C for 12 h, with a heating speed of 0.3 °C min$^{-1}$ and two intermediate dwells at 400 °C and 650 °C for 8 h. Afterwards, they were slowly cooled inside the furnace. The samples were then manually ground, cold-pressed, sealed in evacuated quartz ampoules and heated again in the same conditions. Finally, the samples were once more manually powdered, a $\sim$30 wt% excess of S was added, and $\sim$0.6 g of the resulting powder was charged in a high-density graphite mould that was used in the hot-pressing procedure. The hot-press was made under inert atmosphere (Ar), increasing the pressure at 3 MPa min$^{-1}$ up to 56 MPa and the temperature at 25 °C min$^{-1}$ up to three different dwell temperatures, 700 °C, 720 °C and 750 °C, staying there for 1 h 30 min. Temperature was then decreased to <100 °C at 25 °C min$^{-1}$ and the pressure removed at 3 MPa min$^{-1}$.

Part of each pellet was manually ground and characterized by powder X-ray diffraction (XRD). A PANalytical X'Pert PRO diffractometer (Bragg-Brentano geometry, Cu K$\alpha$ radiation) was used. The powders were placed in a low-noise Si single crystal XRD holder and 2$\theta$ was scanned from 20° to 90°, with a step size of 0.033° and a time per step of 50 s. Phase identification was made through comparison of the collected diffractograms with reference patterns taken from the literature. Cell parameters and theoretical density were calculated and refined from the powder diffraction data, using the Unit-Cell software (*Holland & Redfern, 1997*). Experimental values of density were determined by the Archimedes method. Porosity was estimated by image analysis, using the ImageJ software.

Optical microscopy, scanning electron microscopy (SEM) and energy-dispersive spectroscopy (EDS) were used for microstructure characterization and chemical composition analysis. It was used an optical microscope ZEISS SteREO Discovery V20 and a JEOL JSM-7001F field emission gun scanning electron microscope (accelerating voltage of 25 kV), with an Oxford Instruments EDS spectroscopy system attached.
Thermal stability was evaluated with thermogravimetric analysis (TGA). A Dupont 951 Thermo-gravimetric Analyzer was used. Samples were manually grounded, placed in platinum pans, and heated from 25 °C to 950 °C, at a heating rate of 10 °Cmin$^{-1}$ in an inert atmosphere ($N_2$) flowing at a rate of 60 mL min$^{-1}$.

The nature of chemical bonding was analysed by Raman spectroscopy, using a Horiba LabRam HR Evolution Raman microspectometer (laser with $\lambda = 532$ nm and 10 mW power). Raman spectra were collected from 150 to 1,800 cm$^{-1}$, the laser light being focused with a 100× objective. 4 scans, with 30 s each, were made for each spectrum. Lower laser powers (25–50% of maximum) were required in some measurements to avoid the surface damage.

Electrical transport properties were measured between 20–300 K, at a rate of 0.3 Kmin$^{-1}$ for the Seebeck coefficient and 0.5 Kmin$^{-1}$ for electrical resistivity, using a closed-cycle cryostat. A system based on the Chaikin's device to measure organic single crystals (*Chaikin & Kwak, 1975*) was used to measure the Seebeck coefficient. The samples were first shaped to a needle-like geometry ($\sim$0.5 × 0.5 × 3.5 mm$^3$) and glued with GE varnish to two gold foils (located in two single crystal quartz blocks heated independently), and the foils are glued with GE varnish to the quartz blocks, so that each side of the sample is thermally anchored to one of the blocks. Two gold wires connected to the sample, then establishing the electrical contacts. The voltage was measured with a low frequency AC technique, with a maximum temperature gradient in the sample of 1 K, controlled by two Au-Fe-Chromel thermocouples connected to the quartz blocks. The electrical resistivity was measured in the same bar-shaped samples through the four-point technique, using an AC resistance bridge and a current of 1 mA. Activation energies were obtained from the electrical resistivity data.

### Band structure calculations

The band structure and density of states of $NiS_2$ were calculated with the help of the WIEN2k package (*Blaha et al., 2018*). Calculations were performed within the density functional theory (DFT), using linear augmented plane wave (LAPW) method to solve the Kohn–Sham equations. Lattice parameters and atomic positions were taken from experimental data (*Villars & Calvert, 1986*). Both local spin density approximation (LSDA) and generalized gradient approximation with a modified Becke-Johnson potential (GGA+mBJ) were used to approach the exchange–correlation energy (*Koller, Tran & Blaha, 2012*). The parametrization developed by Perdew-Burke-Ernzerhof was applied for the generalized gradient approximation (PBE-GGA) (*Perdew, Burke & Ernzerhof, 1996*). A cut-off energy of 6 Ry and 1,000 k-points in the irreducible part of the Brillouin zone were used for the self-consistent calculations. The criteria of convergence was set at 0.0001 Ry.

## RESULTS AND DISCUSSION

The powder X-ray diffraction results always point to single phase samples, both after solid-state reaction and hot-pressing (Fig. 1). All peaks are indexed to the $NiS_2$ crystal structure, of cubic Pa3 space group. The lattice parameter after solid-state reaction is

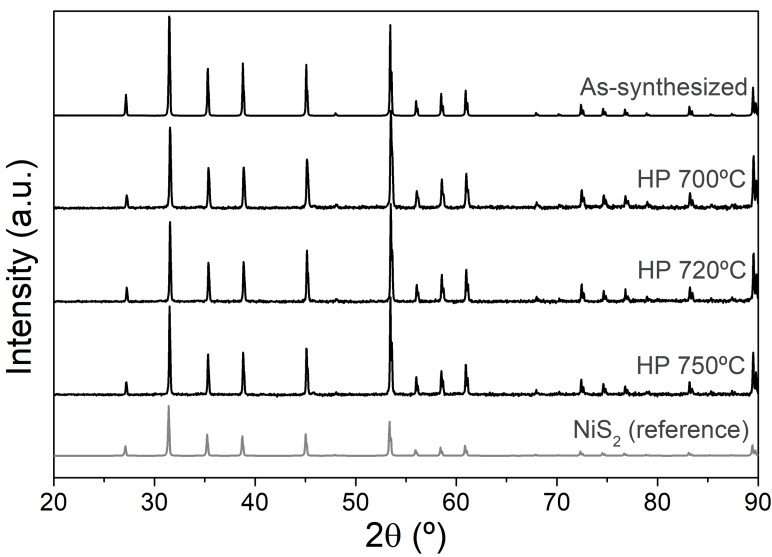

**Figure 1** **XRD diffractograms of as-synthesized NiS₂ and hot-pressed at different temperature conditions.** Simulated pattern of NiS₂ in gray (data taken from *Villars & Calvert, 1986*).

**Table 2** **Comparison of physical properties of NiS₂, at different consolidation conditions.**

| Consolidation conditions | Porosity (%) | Lattice constant (Å) | Theoretical density (g/cm³) | Apparent density (g/cm³) | Relative density (%) |
|---|---|---|---|---|---|
| HP at 700 °C/56 MPa | 6 ± 1 | 5.687(6) | 4.435 | 4.18 ± 0.01 | 94 |
| HP at 720 °C/56 MPa | 6 ± 1 | 5.686(4) | 4.437 | 4.20 ± 0.07 | 95 |
| HP at 750 °C/56 MPa | 4 ± 1 | 5.687(8) | 4.434 | 4.31 ± 0.02 | 97 |

$a = 5.685(2)$ Å. Lattice constants were also calculated for the pellets densified at different consolidation conditions (Table 2), remaining unchanged.

The pellets obtained after the initial heating cycle were highly porous and easily disaggregated, being unsuitable for the electrical transport properties measurements, in agreement with the previous results (*Krill et al., 1976*; *Matsuura et al., 2000*). Representative microstructures of NiS₂, as-synthetized and hot-pressed, were captured by SEM (Fig. 2). As-synthetized samples have several pores at the surface, visible to the naked eye, and poorly agglomerated grains. On the other hand, after hot-pressing there is no distinguishable grain boundaries and the porosity decreased substantially (only small closed pores are present) indicating a successful sintering of the grains. The measured compositions, analyzed by EDS, as well as secondary phases detected, are summarized in Table 3. The microstructure of all samples is homogeneous, being mainly composed of NiS₂. In samples consolidated at 700 °C/56 MPa and 750 °C/56 MPa there is the presence of a sulfur deficient phase, $Ni_{1-x}S$, but in minor amounts (<1vol%). A small deviation from the nominal composition was observed in all samples.

The relative density of the consolidated samples increased with increasing hot-pressing temperature, achieving 97% of the theoretical density when processed at 750 °C and

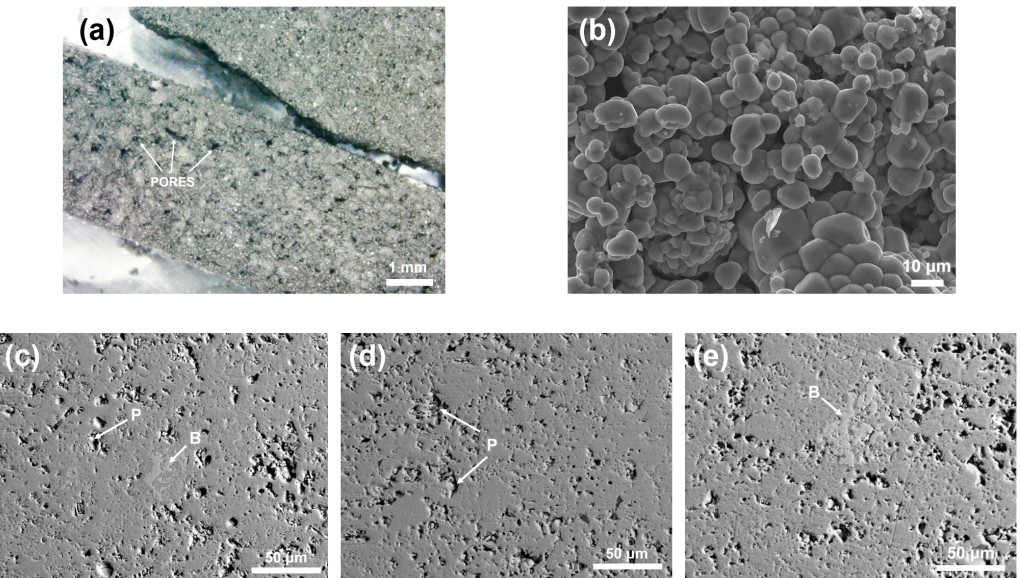

**Figure 2  Optical and SEM images of as-synthesized NiS₂, captured by (A) optical microscope (20×) and (B) SE-SEM (1,000× magnification).** BS-SEM (500×) images of NiS₂ hot-pressed at 56 MPa for 1, 5 h, at different temperatures: (c)700 °C, (b) 720 °C, (c) 750 °C. P –Pores, B –Sulfur-deficient phase $Ni_{1-x}S$.

**Table 3  Measured composition and secondary phases detected by EDS.**

| Consolidation conditions | Measured composition | Secondary phases |
|---|---|---|
| HP at 700 °C/56 MPa | $NiS_{2\pm0.06}$ | $Ni_{1-x}S$ ($x = 0.11$) |
| HP at 720 °C/56 MPa | $NiS_{2.05\pm0.02}$ | Not detected |
| HP at 750 °C/56 MPa | $NiS_{2\pm0.05}$ | $Ni_{1-x}S$ ($x = 0.12$) |

56 MPa. The high relative densities, >94%, and low estimated porosity, <6% (obtained by image analysis), indicate a successful consolidation of the pellets.

Raman spectroscopy was used to characterize the vibrational frequencies specific of the chemical bonds on the hot-pressed samples. Due to their similarity, only one spectrum is shown (Fig. 3). The spectra are in qualitative agreement with literature reports (*Marini et al., 2011*). Vaesite has five Raman active modes: $A_g$, $E_g$ and three $T_g$ modes. Since Ni atoms are located at the center of inversion, all Raman active modes correspond to displacements of the sulfur atoms. Two $T_g$ and $E_g$ modes correspond to S-S pairs libration. $A_g$ and a $T_g$ modes correspond, respectively, to in-phase and out-of-phase stretching of the S-S dimers (*Marini et al., 2011*). In the collected spectra only four peaks were detected: two peaks at 268 and 278 cm⁻¹, corresponding to $T_g(1)$ and $E_g$ symmetries (S-S libration); a peak at 474 cm⁻¹ and a shoulder at 485 cm⁻¹, corresponding to $A_g$ and $T_g(2)$ vibrational modes (stretching vibrations). The fifth Raman active mode, $T_g(3)$, is not visible in the spectra and has never been reported before in previous Raman data available in the literature (*Marini et al., 2011*). In all the consolidation conditions, the peaks are located at the same
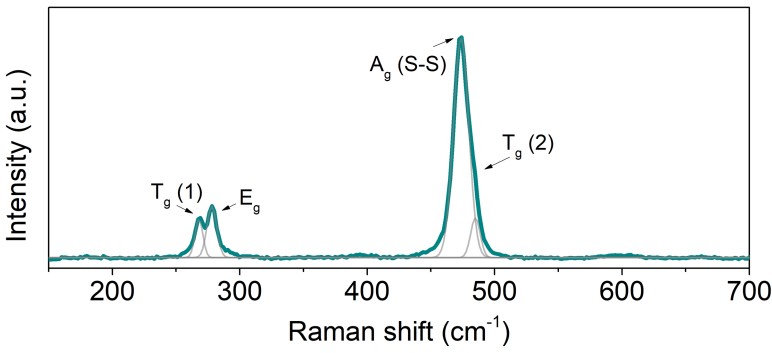

**Figure 3** **Raman spectrum of hot-pressed NiS₂.**

shift values and have similar widths, suggesting that the different sintering temperatures do not introduce distortions or strains in the crystal lattice and that the type and number of bonds are similar.

The thermal stability of $NiS_2$ was evaluated by thermogravimetric analysis under $N_2$ atmosphere, before and after hot-pressing. The results are shown in Fig. 4. There is a small mass loss at ∼80 °C for both non-consolidated and consolidated samples, of 4wt% and <1wt%, respectively, due to dehydration and desorption of chemical species formed during storage under air (the non-consolidated materials was stored for ∼6 months, while the consolidated was characterized just after the preparation). At higher temperatures, a significant mass drop (∼35%) is observed for both samples, due to desulfurization of $NiS_2$. The desulfurization starts at ∼340 ° C in the hot-pressed sample and at ∼440 °C in non-consolidated powders. This difference of almost 100 °C is most likely related with the excess of 30% of sulfur added to the samples prior hot-pressing. The real amount of lost sulfur during the hot consolidation was not controlled and therefore, it is possible that not all the sulfur in excess has been evaporated from the pellet, originating a sulfur-saturated vaesite structure. If that happened, then the early sulfur loss can be caused by the excess of sulfur. Previously reported thermogravimetric analysis of elemental sulfur indicate that sulfur starts to evaporate at 200 °C and by 320 °C the analyzed mass is lost in its total (*Takahashi, Yamagata & Ishikawa, 2015*). If there is excess of sulfur in vaesite structure, then sulfur might start being released at lower temperatures. In order to avoid thermal degradation of $NiS_2$ and formation of S-deficient phases, the service temperature of these materials should not surpass ∼340 °C. There are no previous studies on the mechanisms of decomposition of vaesite but the similarity with pyrite TGA results (*Lambert, Simkovich & Walker, 1998*) suggests that $NiS_2$ might decompose by similar mechanisms of sulfur direct escape from vaesite lattice, followed by a decomposition of $NiS_2$ into $Ni_{1-x}S$ and subsequently, into NiS.

The band structure and density of states of $NiS_2$ calculated using GGA+mBJ (the LSDA give similar results) are shown in Fig. 5. From the DFT calculations, one could expect $NiS_2$ to be metallic due to the partly filled $e_g$ band, in agreement with the previous band structure calculation results (*Vaughan & Tossell, 1983*; *Gibbs et al., 2005*). The temperature

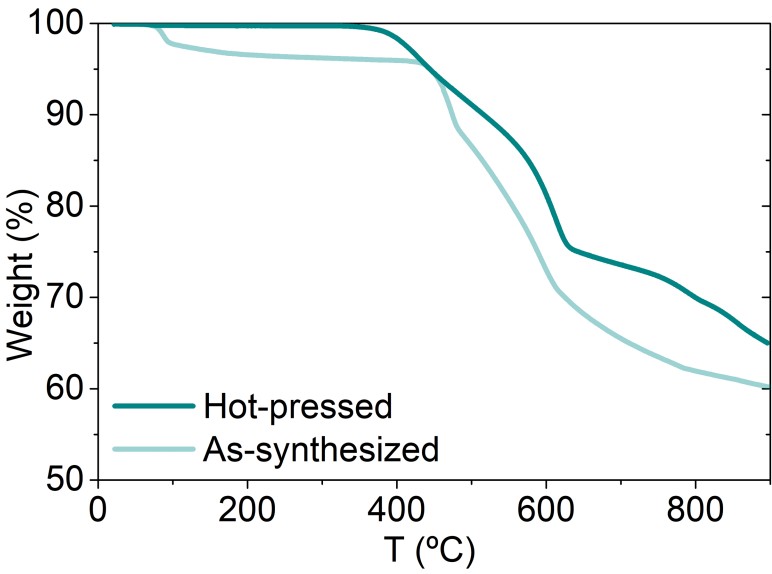

**Figure 4** **Thermogravimetric analysis of as-synthesized and hot-pressed NiS₂.**

dependence of Seebeck coefficient and electrical resistivity, for the different consolidation conditions, are indicated in Figs. 6 and 7. Seebeck coefficient, electrical resistivity and power factors at room temperature are shown in Table 4. In all samples, the Seebeck coefficient is positive, indicating that the major charge carriers are holes (p-type semiconductor). The incoherence between the theoretical and experimental results can be related to the electron–electron interactions that lead to a Mott insulator, i.e., an insulator material due to strong correlation effects originated by electrostatic repulsion between electrons, which are not accounted for by conventional band theories. The bandgap of vaesite has been reported to be 0.27 eV (*Kautz et al., 1972*).

The highest Seebeck coefficient and power factor were obtained for the pellet hot-pressed at 720 °C and 56 MPa. Unlike the other samples, this pellet did not show evidences of secondary sulfur-deficient NiS. No experimental work regarding NiS electrical properties was found, but DFT calculations predict a metallic character (*Persson, 2014*). The presence of a metallic phase, even if in small amounts, is expected to be detrimental to the vaesite thermoelectric properties and can be the reason for the lower power factors on the pellets hot-pressed at 700 °C and 750 °C. Conversely, the non-presence of NiS in the pellet hot-pressed at 720 °C points to a higher sulfur content on it (that due to the small difference to the 1:2 stoichiometry could not be detected), which is able to affect the Seebeck coefficient. Therefore, a higher Seebeck coefficient does not seem to be related with the aggregation of the samples, but with the sensitivity of the electronic properties (carrier concentration and type, conductivity and mobility) to stoichiometric variations and crystal defects (grain boundaries and impurities).

The values of $\rho$ correspond to the values reported in the literature for polycrystalline samples (*Gautier et al., 1972*; *Krill et al., 1976*), being lower than those observed on single crystals and higher than those measured in thin films (see Table 1). A decrease of resistivity

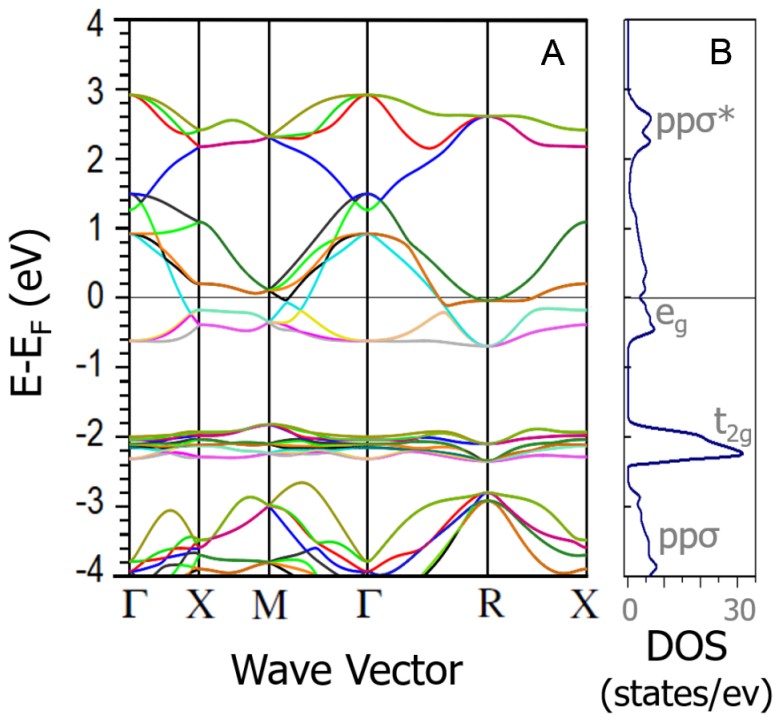

**Figure 5** Band structure (A) and density of states (B) DFT calculations of NiS₂.

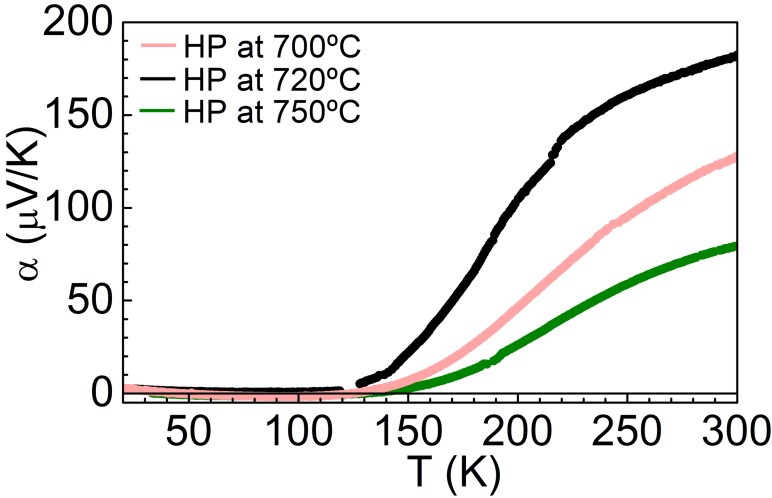

**Figure 6** Temperature dependence of Seebeck coefficient of NiS₂, at different consolidation conditions.

is verified with the increase of the hot-pressing temperature. A higher hot-pressing temperature led to a higher aggregation of the grains, translated into a higher relative density and less grain boundary area, resulting in a decrease of the resistivity. The activation energy of these materials is slightly smaller than the one observed in $Bi_2Te_3$ (Table 4) pointing

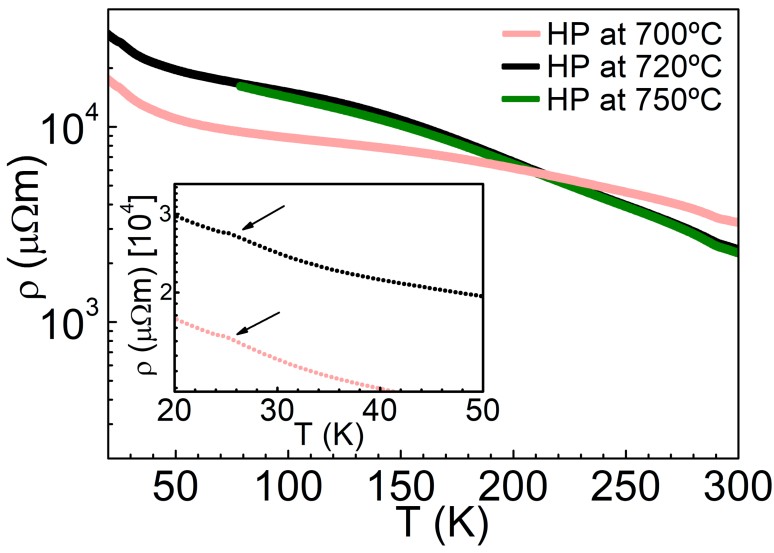

**Figure 7** Temperature dependence of electrical resistivity of NiS$_2$, at different consolidation conditions.

**Table 4** Activation energy ($E_a$), room temperature Seebeck coefficient ($\alpha$), electrical resistivity ($\rho$) and power factor (PF) of NiS 2 hot-pressed at different temperatures.

| Consolidation conditions | $E_a$ (meV) | $\alpha_{300K}$ ($\mu$V/K) | $\rho_{300K}$ ($\mu\Omega$m) | PF$_{300K}$ ($\mu$WK$^{-2}$m$^{-1}$) |
|---|---|---|---|---|
| HP at 700 °C/56 MPa | 43 | 128 | 3,230 | 5.1 |
| HP at 720 °C/56 MPa | 64 | 182 | 2,350 | 14.1 |
| HP at 750 °C/56 MPa | 68 | 119 | 2,257 | 6.3 |

to a possible use as thermoelectric materials close to room temperature. To the best of our knowledge, there are no reported measurements of the Seebeck coefficient of bulk samples. Reported values of room temperature Seebeck coefficient values obtained in this work are of the same order of magnitude of those previously obtained in single crystals (Table 1), which point to a good quality of the prepared materials. Moreover, this good quality is corroborated by the observation of an anomaly in the electrical resistivity data at the spin reorientation temperature, T$_{sr}$~25 K, (insert of Fig. 7), similarly to those observed on single crystals (*Yao et al., 1996*).

The electrical resistivity increases with decreasing temperature, indicating a semiconducting behavior. On the other hand, the decrease of the Seebeck coefficient with decreasing temperature contrasts with the resistivity results and could point to multiple bands, with two types of charge carriers.

## CONCLUSIONS

The preparation of $NiS_2$ by solid-state route, followed by hot-pressing resulted in single phase pellets. Relative densities superior to 94% were achieved. No significant changes in chemical bonds and lattice distortions were verified for the different hot-pressing conditions. Thermogravimetric analysis of these compounds indicates a strong desulfurization above 340 °C, which limits their service temperature. As opposite to the band structure calculations that suggested a metallic behavior, bulk $NiS_2$ is a p-type semiconductor. The maximum power factor obtained for vaesite (14.1 $\mu WK^{-2}$ $m^{-1}$), which is significantly higher than the pristine pyrite (0.06–1.65 $\mu WK^{-2}m^{-1}$) (*Uhlig et al., 2014*) but still far from commercial thermoelectric materials (2,250 $\mu WK^{-2}m^{-1}$) (*Han et al., 2017*), is a good starting point for further improvements. This work indicates that the consolidation conditions had a notable influence on the resistivity, with denser pellets showing a higher electrical conduction, pointing to an intimate relation between the electronic transport properties and the processing conditions, defects (stoichiometric deviations, grain boundaries) and changes in the chemical composition. Therefore, we can expect that, with a proper optimization of the chemical composition and microstructure, these sulfides could become viable thermoelectric materials.

Several aspects were left unexplored in this work. Since the potential of a material for thermoelectricity is also related with its thermal transport properties, a further study of the thermal conductivity is required. The selection of the optimal chemical composition of vaesite, through elemental substitutions, is also necessary. The coupling of these materials in a thermoelectric module also demands good mechanical properties, which so far were never studied. In this project, the thermal stability of vaesite under inert atmosphere was studied but it would be interesting to also evaluate its stability in air (oxidation testing).

### Funding

This work was supported by FCT, Portugal, through UID/Multi/04349/2013, PTDC/EAM-PEC/29905/2017 and M-ERA-NET2/0010/2016 contracts. We also received support from ERA-Net in the framework of the project "THERMOSS", M-ERANET2/0010/2016. There was no additional external funding received for this study. The funders had no role in study design, data collection and analysis, decision to publish, or preparation of the manuscript.

### Grant Disclosures

The following grant information was disclosed by the authors:
FCT, Portugal: UID/Multi/04349/2013, PTDC/EAM-PEC/29905/2017, M-ERA-NET2/0010/2016.
THERMOSS: M-ERANET2/0010/2016.

### Competing Interests

António Pereira Gonçalves is an Academic Editor for PeerJ.

## Author Contributions

- Helena M. Ferreira conceived and designed the experiments, performed the experiments, analyzed the data, prepared figures and/or tables, authored or reviewed drafts of the paper, approved the final draft.
- Elsa B. Lopes, Luís M. Ferreira, Maria H. Casimiro, Luís Santos and Manuel F.C. Pereira performed the experiments, analyzed the data, contributed reagents/materials/analysis tools, authored or reviewed drafts of the paper, approved the final draft.
- José F. Malta analyzed the data, performed the computation work, authored or reviewed drafts of the paper, approved the final draft.
- Antonio Pereira Gonçalves conceived and designed the experiments, analyzed the data, contributed reagents/materials/analysis tools, authored or reviewed drafts of the paper, approved the final draft.

## Data Availability

Raw data are available as Supplemental Files.

## Supplemental Information

Supplemental information for this article can be found online at http://dx.doi.org/10.7717/peerj-matsci.2#supplemental-information.

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
