# Peer review of "Preparation, thermal stability and electrical transport properties of vaesite, NiS$_2$"

_PeerJ Materials Science, doi:10.7717/peerj-matsci.2_

## Round 0.1 · original submission · Minor Revisions

The above manuscript has been reviewed by our reviewers. Comments from their reports appear below.

Please accompany any resubmittal by a summary of the changes made and a brief response to all comments.

Reviewer 1 ·

Basic reporting

Helena et al. reported the transport properties of NiS2 densified by the Hot-press method. They also compared the results with the band calculation. It is important work to clarify the intrinsic bulk properties in NiS2 with large sample dependences. I will be agreement with the publication in PeerJ, if you revised below points.

1) If the values of Seebeck coefficients, resistivity or power factor were already reported in thin film or single crystal study, the authors should note them and compare with those in your hot-pressed NiS2.

2) NiS2 is known to show magnetic ordering at 30-40 K. It is not clear that your hot-pressed samples exhibit a magnetic transition in the temperature dependence of resistivity in figure 7. Since the sharpness of the anomalies by transition may be closely associated with the quality of your samples, authors should show the resistivity data around transition temperature, more clearly, for example by using the enlarged view.

Experimental design

Totally, experimental procedure is well explained in detail in this manuscript. I just think that the authors need to mind below one point.

Before hot-press, the authors synthesized the NiS2 samples (as grown samples). What is the properties of your as-grown sample? I recommend to show the properties for comparison, for example, lattice constant, compositions and electrical resistivity (or the activation energy) of your as-grown sample.

Validity of the findings

1) The authors showed the band calculation of NiS2 with metallic band structures. I am not sure that this work by authors is the first report. If it was already reported, the authors should explain the originality of your band calculation.

2) The authors described that the NiS2 is good starting points for a new family of thermoelectric materials. However, it is not evidently, for readers, that the values of Seebeck coefficients and power factor in your hot-pressed NiS2 are large or not. I suggests these values of typical examples of thermoelectric materials are described in the introduction.

Reviewer 2 ·

Basic reporting

meet the standards

Experimental design

Original research with reasonable technical level.

Validity of the findings

Fig. 6 and Fig. 7 showed that the samples sintered by different temperatures exhibited different transport properties. It should be further confirmed and given explanation.

Additional comments

This is a new phase and the transport properies are worthy to be studied more deeply.

---

## Round 0.2 · accepted · Accept

Thank you for your revisions and rebuttal. We are pleased to inform you that your manuscript has been accepted for publication in PeerJ.